# Microleakage Analysis of Different Bulk-Filling Techniques for Class II Restorations: µ-CT, SEM and EDS Evaluations

**DOI:** 10.3390/ma14010031

**Published:** 2020-12-23

**Authors:** Vincenzo Tosco, Flavia Vitiello, Michele Furlani, Maria Laura Gatto, Riccardo Monterubbianesi, Alessandra Giuliani, Giovanna Orsini, Angelo Putignano

**Affiliations:** 1Department of Clinical Sciences and Stomatology (DISCO), Polytechnic University of Marche, 60126 Ancona, Italy; v.tosco@pm.univpm.it (V.T.); flavit94@libero.it (F.V.); m.furlani@pm.univpm.it (M.F.); r.monterubbianesi@pm.univpm.it (R.M.); a.putignano@univpm.it (A.P.); 2Department of Materials, Environmental Sciences and Urban Planning (SIMAU), Polytechnic University of Marche, 60131 Ancona, Italy; m.l.gatto@pm.univpm.it

**Keywords:** resin composite, dental restoration, scanning electron microscopy, microcomputed tomography, X-ray spectroscopy

## Abstract

This study aimed to compare two different bulk-filling techniques, evaluating the internal and external adaptation of class II resin-composite restorations, by analysing the gap formation using microcomputed tomography (µ-CT) and scanning electronic microscopy (SEM) coupled with energy-dispersive X-ray spectroscopy (EDS). Two standardized mesio/disto-occlusal (MO/DO) cavities were prepared in eight extracted human third molars that were divided, according to the filling technique used, in the following two groups (n = 4): BG (Bulk&Go group) and BT (Bulk Traditional group). After universal bonding application, followed by the light curing, all teeth were restored using a bulk-fill composite. Specimens were scanned with µ-CT to evaluate 3D interfacial gaps. Acquired µ-CT data were analysed to quantify the gap formation. Complementary information to the µ-CT analysis were obtained by SEM. Thereafter, the chemical composition of tooth–restoration interface was analysed using EDS. The µ-CT analysis revealed gaps formation at the tooth–restoration interface for both the BG and BT groups, while within the restoration, only in the BT group there was evidence of microleakage formation. The scanning electron micrographs of both groups showed that the external marginal integrity of the restoration was preserved, while EDS showed the three different structures (tooth surface, adhesive layer and resin composite) of the tooth–restoration interface, highlighting the absence of gap formation. In both BG and BT, the two filling techniques did not show significant differences regarding the internal and external marginal adaptation of the restoration. To achieve a successful restoration, the clinician could be advised to restore a class II cavity using a single increment bulk-filling technique (BG), thus treating it as a class I cavity.

## 1. Introduction

Dental caries represents one of the most prevalent diseases in the world and remains the main challenge for the clinician [1]. It can be classified by location and extent of the lesions produced by the demineralization and, according to the G.V. Black classification system, a lesion placed in the pits and fissures (grooves) of the occlusal (biting) surface of a tooth is considered a class I lesion, whereas a lesion located on a proximal surface of a posterior tooth is considered a class II lesion [2]. The standard treatment for a carious lesion consists of the removal of the infected tooth structure, followed by its restoration using a dental material. Resin composites have become the gold standard in restorative dentistry for their primary role in guaranteeing excellent aesthetics and function [3]. Despite the ongoing research to obtain sophisticated composites with optimal characteristics, these materials may fail predominantly due to their occlusal wear or the insurgence of a secondary caries, due to microleakage that potentially provides loss of marginal integrity [4,5]. Indeed, the polymerization shrinkage stress (PSS) and the deriving microleakage still represents the most significant complication after composite restoration. PSS decreases the adhesive bond strength, which leads to negative clinical implications such as post-operative sensitivity, enamel cracks, and marginal gap formation, ultimately leading to the development of secondary caries [6,7]. Moreover, several studies [8] have confirmed that the cavity configuration (C-factor), known as the ratio between bonded and unbonded surfaces of the composite restoration, plays a crucial role in PSS. The C-factor is an important variable that must be considered when placing composite materials inside the cavity [9]. Cavities with a high C-factor and those with large dimensions have increased PSS, and consequently reduced bond strength. Indeed, especially in large class II cavities with dentin and cementum margins [10,11], despite a lower C-factor compared to class I cavities, a potentially deleterious effect on marginal integrity and gap formation can occur [8]. In order to limit PSS, incremental filling techniques were introduced, whereby resin composite layers are built-up in increments and then cured separately [12]. While the incremental filling technique reduces PSS, it also causes problems such as voids formation among layers and extends the time required to place restorations [13]. In the attempt to decrease the microleakage and shorten the working time, a new generation of resin composites with a novel monomer content and enhanced curing properties, known as bulk-fill composites (BFC), have been recently introduced. BFC have gained success among clinicians for their user-friendly application, especially for posterior restorations [14]. BFC can be cured in a maximal increment thickness of 4 mm with limited shrinkage, thus allowing clinicians to fill the cavity in one single step. This technique, called bulk-fill technique, lets to simplify the procedure with a satisfactory cavity adaptation, allowing to complete the restoration decreasing the chair-time needed, also assuring adequate physical surficial properties [15,16]. Although there are clear clinical and scientific evidences of the advantageous properties of BFC [12,17], more investigations of their mechanical and clinical performances are needed, particularly with regards to PSS and the internal adaptation of the restoration. 

The measurement of the shrinkage in a cavity is technically challenging; thus, several in vitro methods have been described in the scientific literature to analyse the PSS effect [18]. The evaluation of leakage in class II restorations is a common procedure of analysing the long-term stability of adhesive and composite resin systems [19]. The most frequently used method to examine the internal adaptation is the dye penetration, which evaluates the interface between the resin composite and the dental substrate, using basic fuchsin, methylene blue, erythrosine, silver nitrate, or radioactive markers [20]. However, in performing this technique, the samples need to be sectioned, which obviously limits further material characterization and may lead to false interfacial leakage. The internal adaptation can be also evaluated by scanning electron microscope (SEM) after sectioning, obtaining a qualitative evaluation of the surface morphology. Moreover, the energy dispersive X-ray spectroscopy (EDS) is used to investigate the chemical composition of the tooth–restoration interface. Therefore, the effort of coupling these analytical techniques represents a valuable tool to obtain a comprehensive overview of the morphological different structure of tooth/material interface. However, this method is very sensitive, and has several limitations concerning the quantitative assessment, which is based on the operator’s visual evaluation and does not represent, being two-dimensional (2D), the entire gap volumes. 

To overcome all these drawbacks, X-ray micro-computed tomography (μ-CT) has recently been introduced to evaluate both PSS and the internal and external adaptation of resin composites [21,22,23]. Due to the penetrating capacity of X-rays, μ-CT allows clinicians to accurately analyse the restoration without sectioning the samples. Furthermore, it allows to examine internal aspects, irrespective of a sample’s shape or dimensions [24]. Indeed, the μ-CT data can be reconstructed into three-dimensions (3D) to obtain, in a non-invasive and non-destructive way, qualitatively and quantitatively valuable information [25], for instance providing accurate information on the gap formation caused by air bubbles [21]. 

Therefore, the aim of this study was to evaluate and quantify marginal gap formation, air bubbles, and microleakage at the tooth–restoration interface, analysing two different bulk-filling techniques in class II restoration, by means of an innovative experimental protocol based on the combined use of µ-CT, SEM and EDS.

## 2. Materials and Methods

### 2.1. Specimen Preparation

For this study eight sound extracted third molar were collected, from subjects aged between 18 and 30 years. Teeth were surgically extracted for orthodontic reasons at the Section of Stomatology of DISCO Department, Polytechnic University of Marche, Ancona, Italy. According to the Local Ethic Committee guidelines and the 1964 Helsinki declaration, informed consent was obtained from the subjects that were aware that their hard-dental tissues, as discard of the surgical procedures, would be used for research purpose. The inclusion criteria were intact crowns, free of caries, hypoplastic defects, cracks and restorations. Scaler and a hand-scaling instruments were used for surface debridement. Two separated standardized class II slot cavities, mesio-occlusal (MO) and disto-occlusal (DO) were prepared in each tooth (for a total of 16 cavities in 8 teeth) using diamond burs (206/845, Komet Dental, Brasseler GmbH and Co., Lemgo, Germany) on high-speed handpiece (Kavo, Biberach, Germany), under water cooling. The bur was replaced after every five cavities preparations in order to maintain cutting efficiency. The dimensions of the cavities were standardized: Buccolingual width of 4 mm, axial wall of 3 mm from the proximal surface. The cervical margin was placed 1 mm below the CEJ (cemento-enamel junction) (Figure 1a).

All teeth were stored in 0.5% *w*/*w* chloramine solution (NH_2_Cl) at room temperature (25 °C) before and after preparation procedures. After cavities preparation, two different groups including 4 teeth each (n = 8 cavities per group) were formed, according to the filling technique:Bulk&Go group (BG): Each class II cavity was filled using the bulk fill technique (one BFC single increment to the occlusal surface) (Figure 1b);Bulk Traditional group (BT): To begin, the proximal wall was restored in order to transform class II into class I, then the residual cavity was filled using one single increment of BFC (Figure 1c).

All samples, in both groups, were restored using the high viscosity BFC, Filtek One Bulk Fill Restorative (3M ESPE, St. Paul, MN, USA; Table 1). 

Firstly, a circumferential metal matrix (Automatrix MT, Dentsply, Milford, DE, USA) was placed around each tooth, adjusted and fixed around the cavity for a good adaptation. Afterwards, 37% phosphoric acid (3M Scotchbond Etching Liquid, Saint Paul, MN, USA) was applied to the enamel for 30 s (selective enamel etching) and rinsed for 30 s. Specimens were gently air-dried with an air syringe, preventing dehydrating. After that, all groups were treated with Scotchbond Universal bonding, (3M Oral Care, Saint Paul, MN, USA), and “air blowing” for 20 s was accomplished, according to manufacturer’s instructions. Polymerization was carried out by means of the curing lamp LED Elipar (Elipar S10, 3M ESPE, St. Paul, MN, USA) for 40 s with a light irradiance of 1200 mW/cm^2^. The composite material was compacted and adapted, removing the excesses, using the LM-Arte Condensa instrument (LM-Instruments Oy, Parainen, Finland) [26]. Once the restoration was completed, the occlusal surface was finished and polished using abrasive discs (Sof-Lex Discs Coarse and Medium, 3M ESPE, St. Paul, MN, USA), rubber cups (Opti1Step, Kerr Dental, CA, USA), and brushes (Occlubrush Regular, KerrHawe, Boggio, Switzerland); while the interproximal surface was finished and polished using strips (OptiStrip, KerrHawe, Boggio, Switzerland).

### 2.2. Micro-Computed Tomography Analysis

After the restoration, each sample was individually scanned by means of a micro-computed tomography system (μ-CT) Bruker SkyScan 1174 (SkyScan-Bruker, Antwerp, Belgium), installed at the Laboratories of the Research Center and Microscopy Service of Nanostructures (CISMiN), at the Polytechnic University of Marche, Ancona, Italy. Projections settings were as follows: Acceleration voltage 50 kV; beam current 800 μA; aluminium filter of 1 mm; pixel size 9.5 μm and rotation 180° in 0.2° step with an exposure time of 10 s per projection. The average scan time was 5 h. The total number of reconstructed sections for each specimen was approximately 950, obtaining axial information of approximately 9 mm tooth thickness. Tooth projections were converted to cross-sectional slices by NRecon software (Version 1.6.10.2, Bruker, Billerica, MA, USA) with the following correction settings: Ring artefacts (8.0); smoothing (6.0); beam hardening (70%) and proper misalignment compensation. The 3D analysis software VGA Studio MAX (Volume Graphics, vers. 1.2.1, Heidelberg, Germany) was used to perform image analysis and assessment of gap formation within the tooth–restoration interface.

In particular, to evaluate internal adaptation, the *Defect Analysis Tool* of VGA Studio MAX was used. The operation tool was designed to process voxel datasets for internal defects. The parameters quantified were: The number of defects detected at the interface and within the restoration (Nr of defects); the total volume of defects (DV, expressed in µm^3^), the specific surface, i.e., the ratio between the surface and the volume of the defects (DS/DV, expressed in µm^−1^); the mean thickness of the void formation (Mean Thickness, expressed in µm); the volume of mineralized materials (enamel and dentin) and composite (MV, expressed in µm^3^); and the ratio of the latter value to the total volume of the defect (DV/MV, expressed in percentage).

The CTAnalyser software (vers. 1.17.7.2, Bruker Billerica, MA, USA) was employed to create 3D model considering a stack of 200 cross-sectional slices within the defects of interest. 3D reconstructions of enamel, dentin, restorations and gaps, were jointed in a unique colored tooth model by CTvol software (vers. 2.0, Bruker Billerica, MA, USA). 

### 2.3. Scanning Electron Microscopy and Energy-Dispersive X-ray Spectroscopy Analysis

Thereafter, the specimens were fixed in metallic stubs, sputter-coated with gold and observed by a scanning electron microscope (SEM). Scanning electron micrographs were performed for evaluating the presence of gaps and misfits pointed out at the μ-CT (SkyScan-Bruker, Antwerp, Belgium). The images were acquired with a ZEISS SUPRA 40 (CISMiN, Univpm) (ZEISS, Oberkochen, Germany). Firstly, external mesial and distal walls of class II restoration were evaluated at different magnifications. After that, all samples were sectioned according to the axial plane with a diamond blade saw (Buehler IsoMet, Buehler, Manassas, VA, USA) to visualize the internal marginal adaptation and the presence of internal defects of the restoration (at different magnification). For SEM analysis, the surfaces of the sectioned samples were finished and polished with a lapping machine (Buehler Metaserv, Buehler, Lake Bluff, IL, USA) using 600 and 800 grit silicon carbide abrasive papers. The chemical characterization of the tooth–restoration interface was carried out by energy dispersive X-ray spectroscopy (EDS) technique (Bruker, Berlin, Germany), performed by EDAX Element microanalysis. The same samples were used for SEM and EDS. The operating parameters were: 15 mm working distance, 25 kV accelerating voltage at 3000× magnification. The concentrations by weight (%) of the several chemical elements were evaluated.

### 2.4. Statistical Analysis 

The statistical analysis of the morphometric data obtained from the μ-CT was performed using the software package SigmaStat 3.5 (Systat, San Jose, CA, USA). T-tests were performed between the BG and the BT groups on the following parameters: Nr of Defects, DV, DS/DV, Mean Thickness, MV, DV/MV. The statistical significance was analysed at three levels: *p* < 0.05 *, *p* < 0.01 **, *p* < 0.001 ***. All values were expressed as mean and standard deviations.

The statistical power of the experiment was calculated by means of a Post-hoc Power (PhP) Calculator (ClinCalc.com) (Kane SP. Sample Size Calculator. ClinCalc: https://clincalc.com/stats/samplesize.aspx.), setting Continuous Endpoint, Two Independent Sample Study, and α = 0.05.

## 3. Results

The quantification of parameters obtained by internal microleakage analyses using µ-CT on both BG and BT groups are shown in Table 2. Means and standard deviations (m ± SD) for all parameters analysed in BG and BT groups were reported.

The results obtained of various parameters in both groups is represented through the graphs in Figure 2, analysing the data reported in Table 2.

In general, we observed higher standard deviations in the BT than in the BG group for almost all the considered parameters. Moreover, significantly higher specific surface was calculated in the BG group (*p* < 0.001; PhP = 100%) and significantly lower mean thickness (*p* < 0.001; PhP = 99.6%) and DV/MV (*p* = 0.001; PhP = 98.3%) than in the BT group.

SEM analysis displayed the different surface morphology of each sample. Figure 3 displayed the scanning electron micrographs representative of BG and BT samples. Scanning electron micrograph of BG showed that the external marginal integrity of the restoration was preserved (see arrows in Figure 3a). Also, for BT no external gap formation was observed (arrows in Figure 3d).

The combined analysis of SEM and µ-CT data well displayed internal defects, as shown in virtual transversal sections of the restorations performed in the two BG and BT groups (Figure 4 and Figure 5). 

Figure 4 included the 3D model realized by µ-CT (c) and the scanning electron micrographs (a,b) of the internal marginal adaptation of BG restorations. In particular, insets (a) and (b) highlighted the preserved integrity seal, not showing the presence of gaps at the tooth–restoration interface.

On the other hand, Figure 5 showed in both the 3D model (c) and the scanning electron micrographs (a,b), the preserved internal integrity seal with void formation inside of BT restorations.

In all groups, the chemical composition of the tooth–restoration interface was analysed using EDS. Figure 6 reported the levels of the chemical constituents (element and weight in percentage) for the different selected areas, as follows: Selected area 1, corresponding to the enamel surface (E); Selected Area 2, Adhesive layer (Ad); Selected Area 3, BFC surface (BFC).

## 4. Discussion

Marginal adaptation and microleakage formation represent the most common problems associated with composite materials, thus influencing the longevity of dental restorations [19,27]. Such defects can result from inadequate adhesion at the interface of the restoration, due to the following factors: Polymerization shrinkage, degradation of adhesive components because of insufficient light curing, different thermal expansion coefficients between the dental substrate and the composite, finishing and polishing procedures [28]. Nedeljkovic et al. claimed that margins of the class II are, among the restorations, those with higher incidence of secondary caries formation [29]. Moreover, when restorations are placed below the CEJ, the quality of the marginal integrity is uncertain [30]. Findings from the current research indicate that different filling techniques can affect gap formation on the external interproximal margins of class II restorations [23]. The present study focused on the evaluation of internal and external gap formation using two different filling techniques for restoring class II restorations, by means of BFC. A digital non-destructive full analysis of post curing restoration was executed in three dimensions, by using μ-CT. Indeed, μ-CT can be considered a proper method to evaluate BFC, testing new filling techniques and cavity preparations, in an effort to obtain a more favourable C-factor, avoiding PSS and marginal gap formation. Moreover, due to its 3D imaging capabilities, the μ-CT represents the preferred non-destructive method to quantify microleakage, obtaining accurate results on PSS, in agreement with those obtained by dye penetration [31]. The accuracy of the μ-CT 3D imaging has been also shown to allow the visualization of bubbles, which cannot be rendered by conventional methods, such as replica techniques [32].

Unlike other published studies [23], using optimized settings, it was possible to maximize the contrast between the densities of the different enamel, dentin and composite resin materials, in just a single scan, so that it was not necessary to perform scans before and after restoration, saving experimental time and vanishing possible artefacts introduced by the synchronization process. The Defect Analysis Tool of VG Studio MAX software has been calibrated to detect internal defects with a minimum volume of 5000 µm^3^, which means minimum linear dimension of around 20 µm. This threshold was chosen to detect in detail also the smallest possible gap formation; indeed, recent studies observed that a gap of about 60 µm may lead to the development of a wall lesion and hence, predispose postoperative sensitivity and secondary caries [22,23,29,33]. Furthermore, noteworthy is the study of Kuper et al., highlighting that threshold for secondary wall lesion may occur in the range of about 0 to 70 μm [34]. Most of defects were observed inside and in the marginal walls of the restoration, as also reported by previous studies [35,36,37,38,39] and, to a lesser extent, they were found on the mesiodistal and buccolingual walls of restorations. This finding is not completely in agreement with the results of previous studies, which correlated greater contraction in free surfaces (occlusal walls) than in the remaining walls [35,38].

Regarding the number of the total evident defects (Figure 2a), it can be observed that the average value of both BG and BT groups are similar; however, the SD varies greatly, passing from a unit value in the case of BG to 11.5 in the case of BT. This relevant mismatch should be emphasized, as it indicates how the BG technique has more reproducible results regardless of the operator experience and skill.

The characteristics of the observed defects were defined by the following three overall parameters: The specific surface of the defect (DS/DV), the average diameter of the defects (Mean Thickness), and the ratio between the total volume of defects and the absolute volume of the material (DV/MV). For each of them, the post-hoc power analysis clearly showed that the sample size was more than sufficient to support our conclusions. Regarding the morphometry of the defects, by analysing the impact of these parameters individually, it can be observed that the mean thickness (Figure 2c) shows average values strongly in favour of the BG technique, as they are significantly lower and with smaller standard deviations. This presence of smaller gaps in BG samples, combined with the data referred to the total number of defects, leads to a lower total volume of defects. These data are included in the evaluation of the DV/MV parameter, expressed as a percentage (Figure 2d), and it is still clearly in favour of the BG group, with values up to 10 times lower than the BT group. However, μ-CT revealed the presence of bubbles inside the restorations of the BT group, possibly responsible of the higher DV/MV ratio.

The defect specific surface (DS/DV), defined as the surface area per unit volume, is a morphometric parameter that remarkably contributes to the study of microleakage and bubbles formation during restorations. This measure is strongly dependent from the defect’s shape. Interestingly, we found that the defect specific surface (DS/DV) parameter revealed fundamental differences between the two groups. In the BG group, the mean value is higher than in BT group, thus implying a more oriented defect shape (like a layer); conversely, the BT lower values of specific surface indicate the presence of defects with spherical shape, i.e., the bubbles found inside the restorations. Indeed, SEM and µ-CT images confirm these findings, thus displaying, for the BG group, exclusive presence of defects with a laminar shape (Figure 4), while for the BT group, also there is presence of the bubbles with sphere-like-shapes (Figure 5). The laminar shape observed for BG is correlated to defects extended to the cavity wall; while for BT, the spherical shape of the defect inside the restoration is correlated to the possibility of air bubbles entrapment into the restoration. This fact is explained by the different cavity filling techniques, in which the application of different layer (BT) involves certain risks such as the air bubbles incorporation or contamination between layers [40]. 

A limitation of the µ-CT technique is that materials without sufficient radiopacity and with low or no filler content, as for dental adhesives, are difficult to be visualized through the software reconstruction [41,42]; consequently, discrimination between adhesive and air is challenging by means of this method [43]. Therefore, analyses based solely on µ-CT data are not sufficient to definitely state if defects highlighted on the internal margin are really constituted by gap formations or are the areas filled with the adhesive material. In this context, SEM analysis allowed to evaluate, at higher magnifications than µ-CT, those areas where the µ-CT had indicated a hypothetical gap formation. In fact, from a careful SEM visualization in the axial plane of samples, no real gaps were found at the interface, because most of gaps detected by µ-CT analysis were attributable to adhesive layer with a thickness around 40 µm [44]. Therefore, in both techniques (BG and BT) there were not found relevant formations of defects on the internal margin, but just the adhesive layers. To confirm these visual data, a chemical analysis has been carried out, which has confirmed the presence of the exact chemical elements of the adhesive system used for the restoration (Figure 6).

Therefore, the results of the present research may clearly support the BG technique. Noteworthy is the relationship between the C-factor and microleakage [45]. When the clinician deals with a class II cavity, it would be useful to firstly restore the interproximal wall, in order to transform the class II cavity into a class I, although it has a higher C-factor. Without transforming the class II into class I (contrary to BT technique), a more favourable C-factor is obtained, and hence, a less possibility of deleterious effects on marginal integrity and gap formation. Moreover, the application of the modern formulation of BFC, which can be polymerized up to a depth of 4–5 mm, also showing superior mechanical and optical properties (resulting in better aesthetics) [46,47,48,49], allows to reduce the application steps, in comparison with the BT technique, thus minimizing possible clinician’s operational errors and chairside time [25].

## 5. Conclusions

In conclusion, the two bulk-filling techniques, BG and BT, do not show significant differences in the internal and external marginal adaptation of the restoration. The amount and distribution of microleakage visualized in the µ-CT images are closely related to the positioning technique of the composite material. As shown by the µ-CT and SEM analysis, in BT there is the greatest probability of incorporating voids into the restoration.

In general, the standard deviation associated to several morphometric parameters was significantly higher for the BT group than for the BG group, indicating that the reproducibility of the BT filling technique has a great variability (which means that every time the technique is reproduced, there could be a different result). This variability depends on several factors as the type of material, the type of cavity and the operator. The proposed BG technique has characteristics that make it a valid alternative to the traditional BT technique, like smaller size of defects and lower ratios between total volume of defects and the volume of the tooth.

These results suggest that in deep class II cavities, the Bulk&Go filling technique can represent a reliable alternative to traditional filling technique, hence, without transforming class II in class I. Moreover, the association of the performed evaluations allows to achieve a robust correlation between 3D reconstruction, imaging and chemical data, providing reliable and objective information both on the morphology and the chemical composition of the analysed samples. Regarding the tooth–restoration interface, this approach can be considered particularly useful, giving the opportunity to evaluate the effect of restorative procedures from different point of view, despite further studies are needed to confirm the results obtained.

## Figures and Tables

**Figure 1 materials-14-00031-f001:**
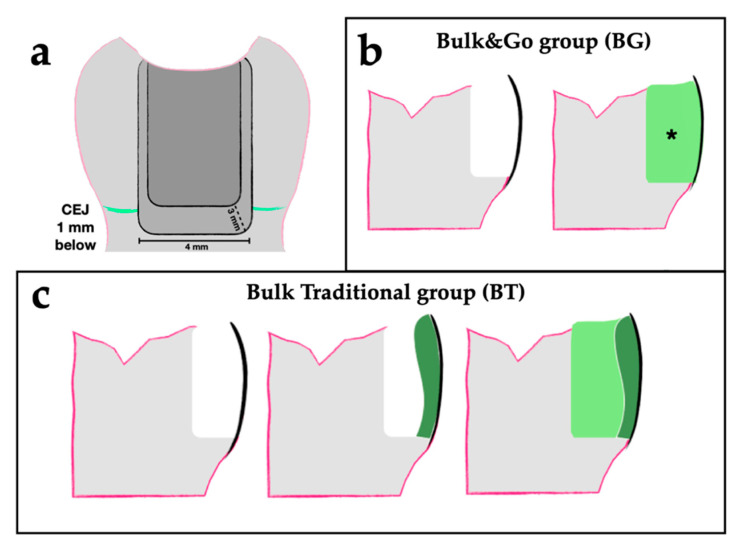
Cavity design and samples preparation. Panel (**a**) represents the stylized cavity preparation of the samples from the mesial view. Panel (**b**) shows the Bulk&Go filling technique: After cavity preparation and automatrix application, only one single increment of BFC was set directly into the cavity, without performing the proximal wall (asterisk). Panel (**c**) displays the Bulk Traditional filling technique: After cavity preparation and automatrix application, the proximal wall was firstly restored, using a bulk-fill composites (BFC), transforming class II into class I (dark green), then the residual cavity was filled using one single increment of BFC (light green).

**Figure 2 materials-14-00031-f002:**
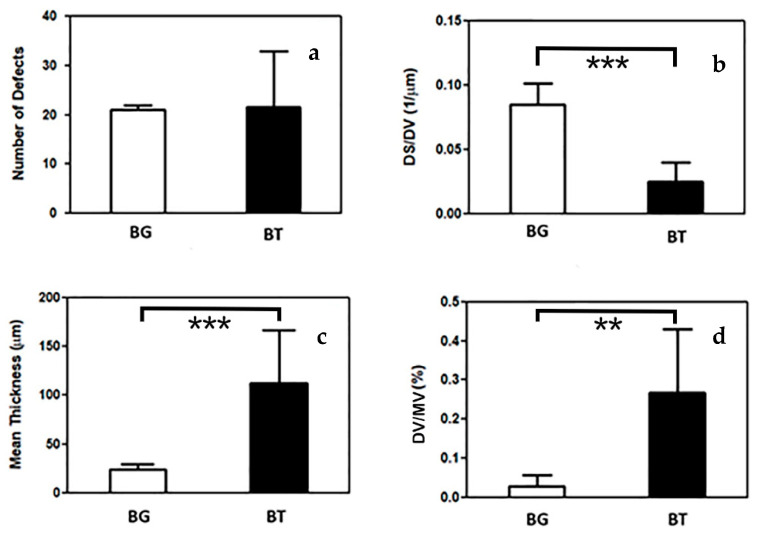
Results obtained by the Defect Analysis Tool, which compared BG and BT groups. As resulting from Table 2, (**a**–**d**) show the number of defects (**a**), DS/DV (**b**), mean thickness (**c**) and DV/MV (**d**). Three levels of statistical significance were considered: *p* < 0.01 **, *p* < 0.001 ***.

**Figure 3 materials-14-00031-f003:**
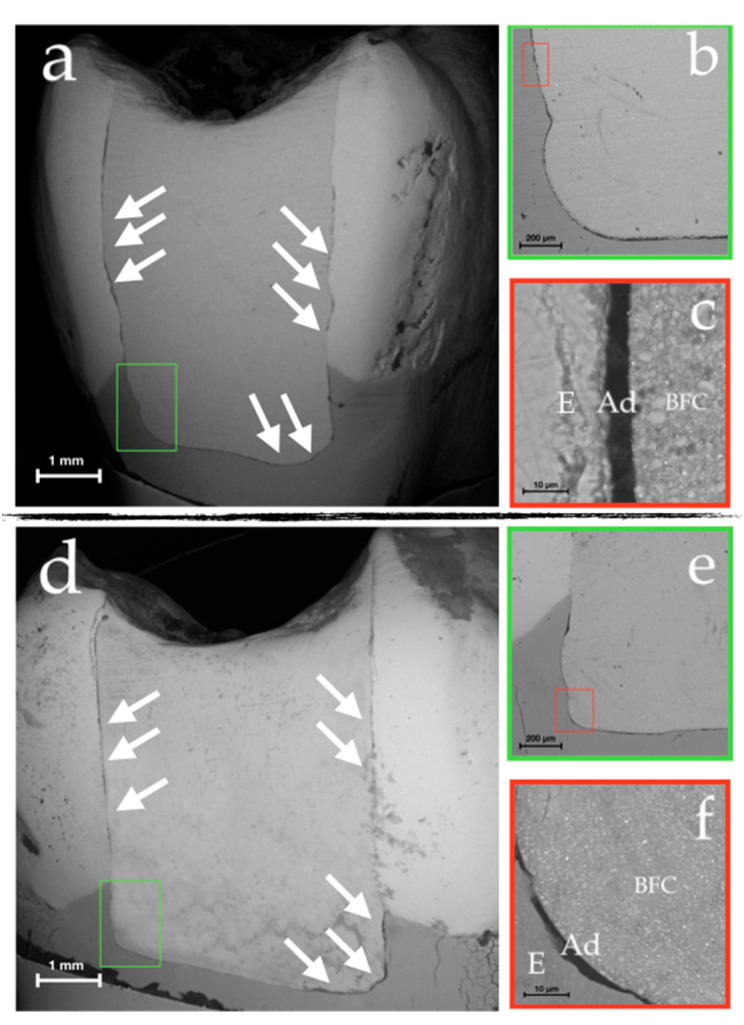
This scanning electron micrographs display a representative mesial view of class II restoration of BG (Bulk&Go group) and BT (Bulk Traditional) sample ((**a**,**d**), respectively). The green inset (**b**,**e**) shows the tooth–restoration interface, which at higher magnification (red inset, (**c**,**f**)) highlights the presence of three different structures: Enamel (E), adhesive layer (Ad) and bulk-fill composite (BFC), without any gap formation on the external margin.

**Figure 4 materials-14-00031-f004:**
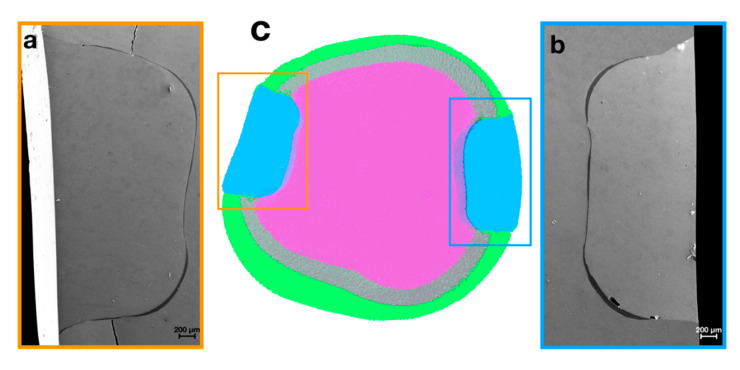
SEM observations of restorations gaps (**a**,**b**) and µ-CT colored 3D reconstruction of a BG representative sample (**c**). Green—enamel; pink—dentin (in transparency the superposition of the dentin with the enamel); blue—restorations.

**Figure 5 materials-14-00031-f005:**
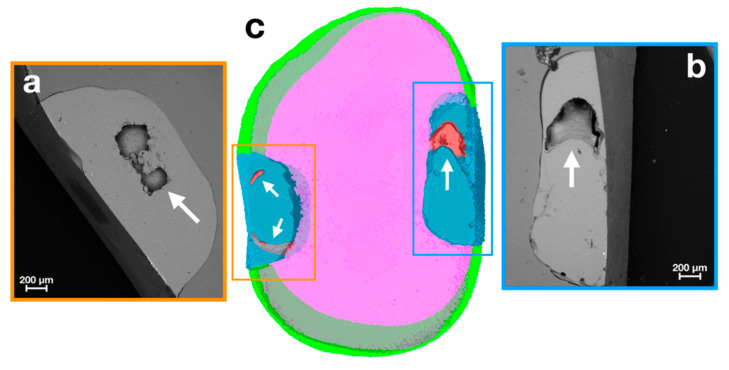
SEM observations of restorations gaps (**a**,**b**) and µ-CT coloured 3D reconstruction of a BT representative sample (**c**). SEM observations of the restorations presenting several defects (arrows in (**a**,**b**)) and µ-CT coloured 3D reconstruction of a BT representative sample with defects (arrows in (**c**)). The void formation in micrograph (**a**) is split in two different defects in the 3D model (orange inset of (**c**)). The void formation in micrograph (**b**) represents the defect inside the restoration (blue inset of (**c**)). Green—enamel; pink—dentin (in transparency the superposition of the dentin with the enamel); blue—restorations (in transparency superposition of the dentin with the restoration); red—defects.

**Figure 6 materials-14-00031-f006:**
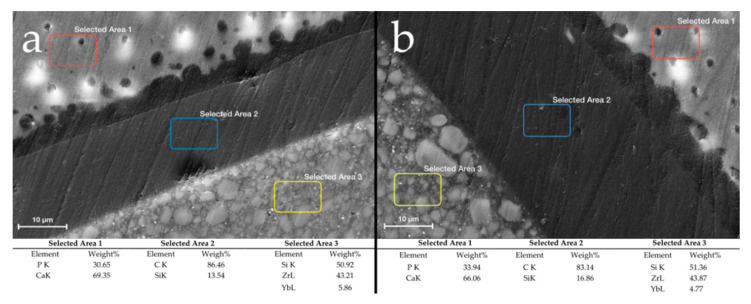
The scanning electron micrographs displayed the three different structure of tooth–interface surface of BG and BT groups ((**a**,**b**) respectively). Selected Area 1: Enamel surface (E); selected Area 2: Adhesive layer (Ad); Selected Area 3: BFC surface (BFC). EDS results of BG and BT groups are reported in the table analysing the different elements as follow: Phosphorous (P), calcium (Ca), carbonium (C), silicon (Si), zirconium (Zr), and ytterbium (Yb).

**Table 1 materials-14-00031-t001:** Composition of the BFC material used in this study.

Material	Manufacturer	Composition
Filtek One Bulk Fill Restorative	3M ESPE, St. Paul, MN, USA	Fillers: Non-agglomerated nanosilica of 20 nm filler size and agglomerated zirconia/silica nanocluster with a size of 5–20 nm. The filler loading was 76.5 wt.% (58.5% by volume). Organic matrix: Bisphenol-A glycidyl dimethacrylate (Bis-GMA) (1–10 wt.%), urethane dimethacrylate (UDMA) (10–20 wt%), triethylene glycol dimethacrylate (TEGDMA) (<1 wt.%), bisphenol A polyethylene glycol diether dimethacrylate (Bis-EMA)-6 (1–10 wt.%), in addition to addition–fragmentation monomer (AFM), aromatic urethane dimethacrylate (AUDMA), and 1,12-dodecane dimethacrylate (DDDMA).

**Table 2 materials-14-00031-t002:** Means ± Standard Deviation o (m ± SD) of data obtained by the Defect Analysis Tool of BG and BT groups. Nr of defects (Number of defects), DV (Defect Volume), DS/DV (Defect Surface/Defect Volume), MV (Materials Volume), DV/MV (Defect Volume/Materials Volume).

Group	Nr of Defect	DV (×10^9^, µm^3^)	DS/DV (µm^−1^)	Mean Thickness (µm)	MV (×10^9^, µm^3^)	DV/MV (%)
BG	21.0 ± 1.0	0.19 ± 0.18	0.085 ± 0.016	24.250 ± 4.969	714 ± 72.7	0.028 ± 0.034
BT	21.5 ± 11.4	2.08 ± 1.09	0.025 ± 0.015	112.032 ± 54.202	829 ± 131	0.268 ± 0.163

## Data Availability

The data presented in this study are available on request from the corresponding author G.O.

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
