# Peer review of "Microleakage Analysis of Different Bulk-Filling Techniques for Class II Restorations: µ-CT, SEM and EDS Evaluations"

_materials, 2020, doi:10.3390/ma14010031_

Round 1

Reviewer 1 Report

The presented work is interesting. It is well supported by literature and has properly selected research methods. I understand that relatively expensive and difficult research techniques were used, but I believe that more repetitions of the experiment would add value to the work. I suggest publishing without changes.

Author Response

Dear Reviewer,

please see the attachement.

Reviewer 2 Report

The simplification of the steps and the adaptation of the restorative materials are critical to clinical success. The result of the study is interesting because the single increment bulk-filling technique showed a comparable level of marginal adaptation to the two-layer one. The results are well presented and supported well the aim of the study. Micro-CT and SEM micrographs are of high quality. The methods of non-invasive evaluation of gap formation are relevant. Therefore, the topic can be of interest to the readership for Materials. The manuscript was carefully written. I recommend acceptance after minor revisions.

Introduction:

L76-101: separate the paragraph into two, one talking about SEM and EDS, another talking about the advantage of the non-invasive evaluation of gap by micro-CT, and introducing the objective of the study.

L94: replace the word “destroying” with other terms such as sectioning.

L112: occlusal-distal change to distal occlusal (DO)

L151: mention the light irradiance of the LED

Methods:

Giving the small sample size (n=4 per group), please indicate the statistical power analysis to determine the minimum sample size.

Minor:

Spelling: L194: Buehler

 Revise references number 23 and 29

Author Response

Dear Reviewer,

Reviewer 3 Report

Introduction section:

  1. In the 2nd sentence, line 41, there is: "it and can be classified..." or some part of the sentence is missing between "it" and "and" or over "and".

Materials and Methods section:

  1. In Figure 1c, according to the drawing the dark green is the external wall that is built first, and the light green is the filling as if it were class I. In the text of the legend it is placed upside down, the light green is the wall and the dark green is the class I type filling. Please check it and put the information in the text correctly.
  2. If from each tooth, 8 in total, you make 16 cavities, why do you put n=4? This part should be explained a little better, as n=4 is a very small and unrepresentative sample size. If 16 cavities are made, shouldn't n=8?

Results section:

  1. In the legend of Figure 2 it says that the results are from Table 4. Please correct this mistake, as there is no Table 4 in the text, and I suppose it is the graphical representation of Table 2, as explained in the text.
  2. Page 7, lines 245-250. Check the text of these short phrases, because it really seems like a repetition and only one of the two phrases may be necessary.
  3. Figure 6 a and b have the same magnifications? Image b looks higher than in a.

Discussion section:

  1. Page 10, paragraph 2, line 316. When you talk about the differences of the standard deviations, saying that the difference could be related to the fact that one technique is more reproducible than the other is something relative. This difference in the standard deviations could also be related to the very small sample size and should be explained somewhere in the discussion, so that it is clear why there is so small sample size.

References section:

  1. Please check the list of references and put them following the template and format of the journal.

Author Response

Dear Reviewer,
